# In Silico Analysis of Temperature-Induced Structural, Stability, and Flexibility Modulations in Camel Cytochrome c

**DOI:** 10.3390/ani15030381

**Published:** 2025-01-28

**Authors:** Heba A. Alkhatabi, Mohammad Alhashmi, Hind Ali Alkhatabi, Hisham N. Altayb

**Affiliations:** 1Faculty of Applied Medical Science, King Abdulaziz University, Jeddah 21589, Saudi Arabia; halkhattabi@kau.edu.sa; 2Hematology Research Unit (HRU), King Fahd Medical Research Center, King Abdulaziz University, Jeddah 22254, Saudi Arabia; 3Center of Artificial Intelligence in Precision Medicines, King Abdulaziz University, Jeddah 21589, Saudi Arabia; 4Department of Medical Laboratory Sciences, Faculty of Applied Medical Sciences, King Abdulaziz University, Jeddah 22254, Saudi Arabia; malhashimi@kau.edu.sa; 5Toxicology and Forensic Sciences Unit, King Fahd Medical Research Center, King Abdulaziz University, Jeddah 22254, Saudi Arabia; 6Department of Biological Science, College of Science, University of Jeddah, Jeddah, 21959, Saudi Arabia; haalkhatabi@uj.edu.sa; 7Biochemistry Department, Faculty of Sciences, King Abdulaziz University, Jeddah 21589, Saudi Arabia

**Keywords:** *Camelus ferus* (wild *Bactrian camel*), *Camelus dromedarius* (*Arabian camel*), thermal adaptation mechanisms, cytochrome c stability and flexibility, thermal titration molecular dynamics (TTMD) simulation

## Abstract

This study investigated the role of cytochrome c in enabling two camel species to endure harsh climates—cold for wild *Bactrian camels* and heat for *Arabian camels*. Cytochrome c is crucial for cellular energy production, and this study examined its behaviour at varying temperatures. Simulations were performed to replicate situations from extremely cold to extremely hot. At lower temperatures, the protein exhibited stability and rigidity, which is appropriate for the frigid habitat of wild *Bactrian camels*. Conversely, at moderate to elevated temperatures, the protein exhibited increased flexibility, enabling *Arabian camels* to sustain their energy processes in arid desert environments. At elevated temperatures, the stability of the protein diminished considerably. The findings of this study demonstrate that cytochrome c has evolved to function efficiently in the specific climates that these camels inhabit. The study’s findings contribute to the development of heat- or cold-resistant enzymes for medicinal applications, as well as insights into animal adaptation to climate change.

## 1. Introduction

Camels are recognized for their exceptional adaptations to severe environmental circumstances. Two species, *Camelus dromedarius* (Dromedary camel) and *Camelus ferus* (Wild *Bactrian camel*), occupy markedly distinct climates [1]. The *Camelus dromedarius* inhabits hot, arid deserts like the Sahara and Arabian Deserts, where average temperatures approximate 303 K (30 °C) and can attain maxima of 320 K (47 °C) [2,3]. The *Camelus ferus* flourishes in the frigid deserts of Central Asia (Gobi Desert), where normal temperatures hover around 280 K (6 °C) but may descend to 245 K (−28 °C) during winter [4]. Cytochrome c, an essential protein in the electron transport chain (ETC), is pivotal to mitochondrial respiration and ATP synthesis. Its optimal operation is essential for energy metabolism during thermal stress. By examining cytochrome c in these camels, this study seeks to comprehend the structural and dynamic adaptations of this vital protein that enable its survival and efficient function under disparate heat conditions [5]. The capacity to endure and adjust to heat stress is crucial for determining the distribution and efficacy of camels [6]. Camels have rapidly evolved genes associated with stress resistance, including those related to DNA damage and repair, apoptosis, protein stabilization, and oxidoreductase activity, with an enrichment in cytochrome c oxidase and monooxygenase activities [7,8].

Cytochrome c is essential in the regulation of body temperature indirectly via its role in cellular respiration and energy production [9]. Cytochrome c is an essential element of the ETC within mitochondria, particularly in Complex III (cytochrome bc1 complex) and Complex IV (cytochrome c oxidase). It enables the flow of electrons between these complexes, facilitating the creation of a proton gradient across the mitochondrial membrane [10]. Cytochrome c plays a central role in energy production by supporting the electron transport chain, where a proton gradient drives adenosine triphosphate (ATP) synthesis [11]. In mammals, tissues like brown adipose tissue use this process for heat generation through non-shivering thermogenesis.

Cytochrome c continues to shuttle electrons, ensuring ongoing heat production during this uncoupling process, which is essential in maintaining body temperature under cold stress [12,13]. Additionally, under cellular stress, cytochrome c can be released into the cytosol to trigger apoptosis. In the context of thermal stress, cytochrome c release is tightly regulated to prevent excessive cell death while adapting to temperature changes. This regulation supports tissue homeostasis and resilience to temperature fluctuations [14]. In thermophilic and cold-adapted species, the structure and function of cytochrome c are often optimized for the prevailing environmental temperatures [15]. Overall, while cytochrome c does not directly regulate body temperature, its function in the electron transport chain and energy metabolism is fundamental to the physiological processes that generate and manage heat. Its efficiency and adaptability under different thermal conditions make it a critical protein in thermoregulation.

Through Thermal Titration Molecular Dynamics (TTMD) simulations, this study aims to understand the structural and functional responses of cytochrome c from *Camelus dromedarius* and *Camelus ferus* across a range of temperatures that reflect the climatic conditions of their habitat. TTMD is a computational technique used to study the behaviour of molecules, particularly proteins, across a range of temperatures. These five temperature were selected accordingly: (i) 245 K (−28 °C): extreme cold in the Gobi Desert [16,17]; (ii) 280 K (6 °C): average temperature in the Gobi Desert [16]; (iii) 303 K (30 °C): average temperature in the Arabian Desert [18,19]; (iv) 308 K (35 °C): close to the physiological temperature of camels; and (v) 320 K (47 °C): extreme heat in the Sahara and Arabian Deserts [20]. Performing TTMD simulations [21] involves systematically varying the temperature during simulations to assess the temperature-dependent structural stability, dynamics, and function of cytochrome c in *Camelus dromedarius* and *Camelus ferus*. This methodology streamlines computer analysis while maintaining biological significance, as modest sequence variations are improbable to affect the protein’s overall structure or function. This approach is ideal for identifying temperature-induced changes in the protein and correlating them with the camel species’ environmental adaptations.

## 2. Materials and Methods

### 2.1. Protein Retrieval

The protein cytochrome c was queried in the Uniprot database for the organism Camelus dromedarius [22], with the ID P68099. A BLAST analysis was conducted to compare the protein sequences of cytochrome c in *Camelus dromedarius* and *Camelus ferus* to examine their thermal stability and molecular adaptations. The FASTA sequence of the protein was obtained using the ID P68099. BLASTp [23] was employed to conduct a pairwise alignment with the cytochrome c of *Camelus ferus*, which was retrieved from the National Center for Biotechnology Information (NCBI) database with the ID XP_032334463.1. The AlphaFold [24] structure of cytochrome c, identified by the ID P68099, was utilized for the Thermal Titration Molecular Dynamics (TTMD) simulation study. This protein’s strong modelling and biological significance led to its use for further analysis to investigate the stability, flexibility, and adaptation of cytochrome c under varying thermal conditions.

### 2.2. Thermal Titration Molecular Dynamics

Thermal Titration Molecular Dynamics (TTMD) simulation is an advanced modelling technique utilized to examine the behaviour and stability of biomolecules, such as proteins, under varying temperature conditions. It assesses whether the original configuration is maintained across a series of molecular dynamics simulations performed at progressively higher temperatures [25]. This study employed TTMD simulations to identify temperature-sensitive areas of cytochrome c. These regions were prioritized for stability. The protein was simulated using TTMD simulation with the GROMACS (GROningen MAchine for Chemical Simulations), a widely used software package for molecular dynamics simulations) 2022.4 package [26,27]. The 3D structure for the protein was transformed using the PDB2GMX module. A cubic simulation box was created, ensuring a separation of 1.0 nm between the protein and the box boundaries. The system was also solvated within the solvation box following the TIP3P model [28]. Subsequently, sodium (Na+) and chloride (Cl-) ions were added to facilitate the ionization of the system. The system was later minimized using the steepest descent method with 50,000 steps to resolve steric conflicts. Furthermore, system stability was achieved with the implementation of bond limitations by the LINCS algorithm [29]. During the equilibration phase, the protein was restricted utilizing the constant number, volume, and temperature (NVT) and isothermal–isobaric (NPT) ensembles. The system’s stability was preserved by controlling temperature through the velocity-scaling approach [30] and pressure using the Parrinello–Rahman coupling method [31]. An NVT ensemble was conducted with a 2 fs timestep, modelling the full system for 100 ps at multiple temperatures (245 K, 280 K, 303 K, 303 K, and 320 K). The identical approach was utilized to maintain the system’s pressure at NPT conditions: 1 ns at 245 K, 280 K, 303 K, and 320 K, at 1 atmosphere. The parameters for the NVT (constant number, volume, and temperature), NPT (constant number, pressure, and temperature), and production runs are shown in Appendix A for clarity and reproducibility. For each simulation case, the reference temperature (ref_t) was set at 245 K, 280 K, 303 K, 308 K, and 320 K. Appendix A shows the summary of the key parameters used. Throughout the 100 ns production cycle, the structure coordinates were recorded every 10 ps. Root mean square fluctuations (RMSFs) and root mean square deviations (RMSDs) were utilized to assess the conformational stability and predictability of the TTMD results. The post-TTMD simulation analysis was performed on the visual platform “Analogue,” developed by Growdea Technologies [32,33] (https://growdeatech.com/Analogue/, accessed on 3 December 2024).).

### 2.3. Principal Component Analysis

Covariance analysis, also known as principal component analysis (PCA) or essential dynamics, can identify connected movements [34,35]. The trajectory was preprocessed for principal component analysis by removing the periodic boundary condition. The covariance matrix was computed with the Gmx_covar module of GROMACS. The covariance matrix illustrates the link between the atomic fluctuations of the protein–ligand complex. The gmx analysis function was employed to compute the eigenvalues and eigenvectors of the covariance matrix. The GROMACS tool ‘gmx anaproj’ was employed to compute the principal component (PC) coordinates for each frame. The regulation was established to demonstrate the progression and distribution of conformational states identified through PCA.

### 2.4. Free Energy Landscape

Investigations into the equilibrium state, defined by the minima on the Free Energy Landscape (FEL), and the transitional state, by the barriers on the FEL, may provide crucial information on biological processes such as biomolecule recognition, aggregation, and folding [36]. In order to calculate the FEL, the energy distribution was approximated using Equation (1).
(1)
$$∆G\left(X\right)=-kBTln\;P(X)$$


*P*(*X*) is the probability distribution of the system along the reaction coordinate, *kB* is the Boltzmann constant, *G* is the Gibbs free energy, and *X* is the reaction coordinate.

### 2.5. Entropy Analysis

Normal Mode Analysis (NMA) in GROMACS serves as a competent method for quantifying entropy variations in molecular systems, especially regarding the dynamics and stability of proteins and other macromolecules [37,38]. MD simulations were conducted with the aim of calculating the protein’s entropy. The computation of entropy by NMA emphasizes the vibrational contributions, offering insights into the stability and flexibility of molecular structures.

## 3. Results and Discussion

### 3.1. Protein Structure

The findings as shown in Figure 1a demonstrate an extraordinarily high sequence identity of 99.05% (104 out of 105 residues), with merely one amino acid alteration (DLI in *C. dromedarius* against DLT in *C. ferus*). This substitution of threonine (polar and hydrophilic) with isoleucine (nonpolar and hydrophobic) occurs in the C-terminal region, a structurally significant area of the protein. This region may play a critical role in maintaining protein stability and flexibility, especially under varying thermal conditions. The hydrophobic nature of isoleucine could contribute to enhanced packing and stability in warmer environments, aligning with the heat-adapted physiology of *C. dromedarius*. Conversely, the polar nature of threonine in *C. ferus* may allow for increased interactions with the aqueous environment, aiding in cold adaptation. These differences highlight the fine-tuned molecular mechanisms that underlie the environmental resilience of camel cytochrome c.

The findings also demonstrate significant conservation of cytochrome c between the two species, highlighting its essential function in the electron transport chain and mitochondrial respiration. Due to the significant similarity, the cytochrome c structure of *Camelus dromedarius* (UniProt ID: P68099) was used as a model for subsequent TTMD simulations. Simulations were conducted at 245 K, 280 K, 303 K, 308 K, and 320 K, temperatures indicative of the extreme cold and heat extremes encountered by *C. ferus* and *C. dromedarius*, respectively. This study aims to elucidate the molecular mechanisms by which cytochrome c adapts to the varying heat environments of different camel species.

The cytochrome c protein structure obtained via AlphaFold, illustrated in Figure 1b, is a colour-coded representation based on per-residue model confidence ratings (pLDDT), ranging from 0 to 100. These scores provide insight into the dependability of the projected structure at each residue. Regions exhibiting pLDDT values ranging from 90 to 100, seen in dark blue, signify exceptionally high confidence, indicating robust and well-structured domains such as alpha helices and beta sheets. Cyan regions, representing scores between 70 and 89, denote strong confidence, suggesting these areas are likely organized but with marginally less certainty than the dark-blue regions. Yellow regions, with scores ranging from 50 to 69, indicate moderate confidence, typically denoting flexible loops or less ordered areas. Ultimately, orange zones, exhibiting scores below 50, signify poor confidence and may correspond to innately disordered regions or areas that lack a well-defined structure in isolation. As major regions had elevated pLDDT scores (blue and cyan), the structure of cytochrome c was used for functional and structural study to investigate stability and essential roles.

### 3.2. Thermal Titration Molecular Dynamics

TTMD simulation and thermodynamic analyses of cytochrome c yield significant insights into the structural and functional adaptations of this protein to extreme thermal environments, mirroring the distinct habitats of *Camelus ferus* (cold Gobi Desert) and *Camelus dromedarius* (hot Arabian Desert) [39,40]. Cytochrome c is a conserved protein essential for electron transport and energy metabolism, and its functionality under varied environmental conditions highlights its evolutionary significance [41,42].

#### 3.2.1. Root Mean Square Deviation

The root mean square deviation (RMSD) study of cytochrome c at different temperatures (245 K, 280 K, 303 K, 308 K, and 320 K) indicates distinct variations in the protein’s structural stability with these settings. Figure 2 shows the RMSD of the protein at different temperatures. At 245 K and 280 K, representative of the cold temperatures characteristic of the Gobi Desert, the RMSD remained continuously low at 0.1–0.2 nm, exhibiting signifying elevated structural stability with little conformational changes. This behaviour corresponds with the adaptation of *Camelus ferus* to cold conditions, where sustaining protein stability at reduced kinetic energy levels is crucial for enzymatic activity. As the temperature ascends to 303 K and 308 K, the RMSD escalates to approximately 0.15–0.3 nm and 0.2–0.4 nm, respectively. This indicates moderate structural flexibility, beneficial for cytochrome c’s activity in the comparatively warmer environments of *Camelus dromedarius*.

At the elevated temperature of 320 K (yellow line), the RMSD demonstrates considerable swings, increasing to around 0.2–0.45 nm, which suggests partial destabilization and heightened conformational dynamics. This indicates that cytochrome c from *Camelus ferus* may have difficulty maintaining stability under elevated heat stress, but *Camelus dromedarius* may demonstrate some tolerance to similar conditions. This comparison underscores the temperature-dependent adaptations of cytochrome c. Here, cytochrome c exhibits remarkable stability in cold conditions, appropriate for *C. ferus*, and shows flexibility at moderate temperatures, optimal for *C. dromedarius*. Extreme heat (320 K) seems to surpass the structural constraints of cytochrome c, highlighting the significance of environmental adaptations in preserving protein functionality among animals inhabiting diverse thermal environments. This dynamic research highlights the molecular foundation of cytochrome c’s thermal stability and its essential function in facilitating camels’ survival in severe environments.

#### 3.2.2. Root Mean Square Fluctuation

The root mean square fluctuation (RMSF) graph for cytochrome c illustrates temperature-dependent residue flexibility, offering insights into its structural dynamics across varying thermal environments. Figure 3 shows the RMSF at different temperatures. At 245 K and 280 K, reflective of the cold habitat of *Camelus ferus*, the RMSF values continuously exhibit low levels over the majority of residues, with only slight peaks detected in loop regions and terminal residues. This indicates the protein’s considerable stiffness and stability in cold settings, which is crucial for preserving its functional integrity in the low-energy environments of the Gobi Desert. The restricted flexibility guarantees that cytochrome c can efficiently engage in electron transfer within the mitochondrial electron transport chain, notwithstanding the diminished kinetic energy linked to low temperatures.

Conversely, at elevated temperatures (303 K, 308 K, and 320 K), flexibility incrementally enhances, especially in loop regions (residues 43–56) and terminal residues (1–5 and 100–105). Here, at 303 K, the moderate flexibility signifies structural adaptability, consistent with the physiological temperature range of *Camelus dromedarius*. During the MD simulation at 308 K, flexibility is notably enhanced, particularly in unstructured regions, potentially promoting effective electron transport in elevated desert temperatures. Further, at 320 K, the RMSF exhibits a pronounced peak in these locations, indicating substantial conformational changes and diminished structural stability. The enhanced flexibility under extreme heat underscores the limitations of cytochrome c’s thermal stability and indicates a possible functional impairment at very elevated temperatures. The RMSF study highlights the adaptive variations in cytochrome c between *Camelus ferus* and *Camelus dromedarius*, with structural stiffness facilitating cold adaption and increased flexibility promoting functionality in warmer climates.

#### 3.2.3. Conformation

The structural analysis of cytochrome c at a variety of temperatures (245 K, 280 K, 303 K, 308 K, and 320 K) reveals temperature-dependent conformational changes that occur after 100 ns of TTMD simulations. The effects of temperature on the protein’s structural integrity and flexibility are explicated in each panel, which juxtaposes the beginning structure (0 ns) with the final structure (100 ns) at a specific temperature.

Figure 4 shows the conformation at 0 ns and 100 ns during the MD simulation at different temperatures. The conformations exhibit significant similarity between 0 ns and 100 ns at 245 K and 280 K, with only minor differences observed in the loop and terminal regions. Cytochrome c’s structural integrity and rigidity are maintained at reduced temperatures, as evidenced by this. Persistently high RMSF are observed in terminal residues, such as Met1, Asn104 and GLU105, which suggests that the protein is functional in all the environments. Significant conformational changes are apparent at 303 K and 308 K, particularly in the loop and disordered regions (e.g., the vicinity of His27, Lys26, and the terminal residues Asn104/Glu105). This increased flexibility suggests that the protein has responded to elevated temperatures by adapting, thereby enabling it to maintain its functional dynamics in the higher temperatures that *Camelus dromedarius* encounters. Notable deviations are observed at the elevated temperature of 320 K, particularly in regions such as Gln43 and Val44, which implies the partial destabilization of the structure. The thermal stability of cytochrome c is limited at elevated temperatures, as evidenced by the substantial structural changes in the terminal residues (Met1, Asn104, and Glu105). The temperature-dependent conformational alterations observed after 100 ns of TTMD simulations, in comparison to the baseline structure at 0 ns, are underscored by the structural analysis of cytochrome c at various temperatures (245 K, 280 K, 303 K, 308 K, and 320 K). The initial conformation (0 ns) and the final conformation (100 ns) are juxtaposed in each panel to assess the effect of temperature conditions on the protein’s structural integrity, stability, and flexibility.

The RMSD values between the 0 ns and 100 ns conformations of cytochrome c at various temperatures indicate the degree of structural rearrangement. At 245 K, the RMSD is 1.08 Å, signifying negligible variation and a rather stable conformation. Conversely, at 280 K and 303 K, the RMSD values markedly rise to 2.35 Å and 2.30 Å, respectively, indicating considerable structural alterations and implying enhanced protein flexibility or destabilization at these temperatures. At 308 K and 320 K, the RMSD values diminish to 1.69 Å and 1.68 Å, indicating mild conformational aberrations, with analogous structural rearrangements noted at these elevated temperatures. These data underscore a temperature-dependent trend in the protein’s structural dynamics, with the most significant alterations happening at around 280 K and 303 K. The areas exhibiting elevated RMSF are marked in purple, signifying regions of significant flexibility within the protein structure. The flexible areas, particularly residues MET1, ASN104, and GLU105, are significant contributors to the observed structural variations at varying temperatures.

The protein exhibits negligible variations between 0 ns and 100 ns at 245 K and 280 K, particularly in the secondary structural components such as alpha helices. Met1 and Asn104, as well as the loop sections and terminal residues, demonstrate remarkable structural stability and rigidity, with minimal alteration. This demonstrates the cold-adapted characteristics of cytochrome c in *Camelus ferus*, which is required to maintain its structural integrity in the Gobi Desert, which is characterized by low temperatures and diminished kinetic energy. The protein’s robustness in cold conditions is underscored by the nearly identical structure at 0 ns and 100 ns, which is consistent with the low RMSD and RMSF values that were observed at these temperatures. A significant amount of conformational change occurs between 0 ns and 100 ns at 303 K and 308 K, particularly in loop regions and surface-exposed residues like His27, Lys26, and the terminal residues (Asn104/Glu105). These changes are indicative of the protein’s adaptive response to the elevated thermal energy characteristic of *Camelus dromedarius*, as they indicate enhanced flexibility and dynamic behaviour. The thermal stability of cytochrome c is constrained at elevated temperatures, as evidenced by the substantial conformational changes observed at 100 ns. This suggests that its functional efficacy may be compromised under these conditions.

The conformational analysis of structures at 0 ns and 100 ns emphasizes the distinctive temperature adaptations of cytochrome c in *Camelus ferus* and *Camelus dromedarius*. The protein’s exceptional structural stability is crucial for cold adaptation in *Camelus ferus* at low temperatures (245 K and 280 K). The observed functional adaptation to the higher climates of *Camelus dromedarius* is suggested by the observed flexibility at moderate temperatures (303 K and 308 K). The protein experiences significant structural changes and decreased stability at temperatures exceeding 320 K, which showcases the limits of its thermal resilience. The structural mechanisms that enable cytochrome c to adapt to the divergent habitats of various camel species are elucidated by many findings.

#### 3.2.4. Radius of Gyration

The compactness of the protein structure during the 100 ns simulation period is demonstrated by the Radius of Gyration (Rg) figure for cytochrome c at various temperatures (245 K, 280 K, 303 K, 308 K, and 320 K). The distribution of atomic distances from the centre of mass is quantified by Rg, with smaller values indicating a more compact structure and greater values indicating loosening or unfolding. Figure 5 shows the Rg of the protein at different temperatures. Rg values are consistently low in the range of 1.30–1.33 nm at 245 K and 280 K, indicating that the protein maintains a compact, densely packed conformation. The structural rigidity and stability of cytochrome c at reduced temperatures are consistent with the climatic conditions of *Camelus ferus*, as indicated by this figure. The protein’s robustness and reduced flexibility in frigid environments are substantiated by the limited variations that have occurred over time. The Rg values exhibit a slight increase to 1.32–1.36 nm in comparison to lower temperatures at 303 K and 308 K. This suggests a significant degree of structural relaxation, which may be associated with the increased flexibility necessary for functional adaptations in high-temperature environments. These temperatures are consistent with the physiological range of *Camelus dromedarius*, and this adaptability may improve the efficiency of electron transport and protein functionality.

The Rg values exhibit the most significant oscillations to 1.33–1.45 nm at 320 K, which are accompanied by a general increasing trend that suggests structural loosening and partial destabilization. This is consistent with the observed trends in RMSD and RMSF at this temperature, emphasizing the structural stability constraints of cytochrome c in the presence of significant thermal stress. This behaviour suggests that cytochrome c may begin to lose its functional integrity in the harsh conditions of the Sahara and Arabian Deserts. In *Camelus ferus*, the compact conformation of cytochrome c exhibits its cold adaptation by maintaining structural integrity despite a reduction in kinetic energy at lower temperatures (245 K and 280 K). The functional adaptations of *Camelus dromedarius* to arid conditions are reflected in the marginal increase in Rg and moderate flexibility at intermediate temperatures (303 K and 308 K). The protein may approach thermal instability at the elevated temperature of 320 K, as evidenced by the increased Rg and greater variations, which emphasizes the maximal threshold of its thermal resilience.

The Rg study emphasizes the distinctive structural modifications of cytochrome c in response to changing temperature conditions. In *C. ferus*, the protein maintains compactness and stability at low temperatures, which aids in cold adaptation. However, in *C. dromedarius*, the protein exhibits significant flexibility at physiological temperatures. The significant relaxation observed at 320 K underscores the challenges that cytochrome c faces when subjected to severe temperature stress, providing a glimpse into its thermal thresholds and potential functional impairments.

#### 3.2.5. Solvent-Accessible Surface Area

The protein’s surface exposure to the surrounding solvent during the 100 ns simulation is illustrated in the Solvent-Accessible Surface Area (SASA) plot for cytochrome c at a variety of temperatures (245 K, 280 K, 303 K, 308 K, and 320 K). SASA is a critical metric for protein stability and folding, as changes in the accessible surface area may indicate structural relaxation, unfolding, or hydrophobicity changes. Figure 6 shows the SASA of the protein at different temperatures. The SASA values are consistently low in the range of 65–70 nm² at 245 K and 280 K (red line), with negligible variations. This indicates a configuration that is densely packed, with hydrophobic residues being exposed to the solvent in a limited capacity. The SASA values are consistent with the compact and stable characteristics of the protein that were identified in the Rg and RMSF study. This further supports the notion that cytochrome c maintains its structural integrity under the cold conditions that are characteristic of *Camelus ferus*. A slight relaxation of the protein conformation is indicated by the modest rise to 66–72 nm² in SASA at 303 K and 308 K. This modification indicates that the protein has adapted to the elevated temperatures of *Camelus dromedarius*, as evidenced by the increased flexibility and accessibility of certain surface residues. The minor increase in SASA is suggested by the tolerable variations, which suggest that the protein’s structure is actively adapting to these intermediate temperatures.

The SASA values exhibit significant fluctuations and a significant increase to 68–75 nm² at 320 K. This indicates a substantial increase in solvent exposure, which suggests that the protein is partially destabilized and that its structure is unravelling due to the extreme heat. The protein’s increased dynamics, particularly in loops and disordered regions, may lead to functional impairment under heat stress, as indicated by the substantial alterations. Cytochrome c maintains a compact conformation with limited solvent exposure at reduced temperatures (245 K and 280 K), thereby increasing its stability in the frigid habitat of *Camelus ferus*. In contrast, the protein demonstrates moderate surface exposure at intermediate temperatures (303 K and 308 K), which is indicative of its functional adaptability in the arid habitats of *Camelus dromedarius*. The structural instability of cytochrome c under extreme thermal conditions is underscored by the substantial variations and pronounced increase in SASA at the maximal temperature of 320 K.

The SASA analysis confirms the distinctive temperature adaptations of cytochrome c in *Camelus ferus* and *Camelus dromedarius*. The protein’s mild relaxation at moderate temperatures enables effective operation in warmer settings, while its stability and compactness are maintained at low temperatures, assuring functionality in cold environments. The protein’s thermal threshold is underscored by the substantial increase in solvent exposure at 320 K, which implies that it may be destabilized and functionally impaired in the presence of severe heat stress. These findings elucidate the molecular basis of cytochrome c’s ability to adapt to a variety of environmental conditions.

### 3.3. Principal Component Analysis

The principal component analysis (PCA) plots for cytochrome c at temperatures of 245 K, 280 K, 303 K, 308 K, and 320 K offer a clear understanding of the protein’s conformational diversity and dominant modes of motion under a variety of thermal conditions. PCA captures large-scale motions and facilitates the visualization of conformational space that is investigated during TTMD by projecting the atomic fluctuations along principal eigenvectors. Figure 7 shows the PCA of the protein at different temperatures.

The compact and densely clustered points in the PCA plot demonstrate that the conformational space investigated by cytochrome c is highly restricted at 245 K. This is consistent with the protein’s stability and frigid adaptation in the Gobi Desert environment for *Camelus ferus*, as it suggests minimal conformational diversity and rigid structural behaviour. The low RMSF and Rg values observed at this temperature are consistent with the reduced atomic fluctuations. A modest increase in conformational sampling is observed in the PCA plot at 280 K in comparison to 245 K, as the clusters become broader. This implies a minor increase in flexibility, despite the protein’s overall stability. This is indicative of the near-physiological conditions in *Camelus ferus*, where cytochrome c maintains its functional compactness when subjected to moderate motion.

The conformational space expands further at 303 K, resulting in a greater dispersion of the clusters along the principal eigenvectors. This suggests that the protein has adapted to the arid climate of *Camelus dromedarius*, as evidenced by its increased flexibility and dynamic behaviour. The expanded conformational range implies that cytochrome c is actively investigating the configurations necessary for optimal performance at elevated temperatures. The conformational space exhibits substantial dispersion at 308 K, as evidenced by the formation of numerous distinct clusters in the PCA plot. This implies that the protein demonstrates a greater degree of conformational diversity, which suggests that it is more capable of adapting to the increased thermal energy. Cytochrome c’s function in the electron transport chain is contingent upon this adaptability in arid environments. The PCA plot at 320 K reveals the most extensive conformational space that has been investigated, with points that are widely dispersed and overlapping clusters. This suggests that cytochrome c is highly adaptable, as it can sample a diverse array of configurations. Nevertheless, the protein’s excessive dispersion implies that it may be on the brink of structural instability, which is indicative of the thermal stress experienced during extreme heat conditions. This is consistent with the high RMSF, Rg, and SASA values that were observed, which suggest a potential functional compromise.

At lower temperatures (245 K and 280 K), the PCA plots reveal restricted conformational motion, indicative of cold adaptation and structural stability in *Camelus ferus*. As the temperature increases to 303 K and 308 K, the conformational space expands, reflecting the flexibility and dynamic behaviour required for *Camelus dromedarius* to function efficiently in warmer climates. At the highest temperature (320 K), the broad dispersion highlights the protein’s structural loosening and potential destabilization under extreme heat. The thermal adaptability of cytochrome c is illustrated by the PCA. In *Camelus ferus*, restricted motions ensure stability at low temperatures, while in *Camelus dromedarius*, enhanced flexibility facilitates functionality at moderate temperatures. Nevertheless, the protein’s adaptive range and potential compromise under extreme heat stress are underscored by the excessive conformational diversity at 320 K, which signals the limits of cytochrome c’s thermal resilience.

### 3.4. Free Energy Landscape

The Free Energy Landscape (FEL) analysis of cytochrome c at different temperatures offers a better understanding of the protein’s adaptability and conformational stability. Figure 8 shows the FEL of the protein at different temperatures. The FEL exhibits energy minima that are firmly confined at 245 K and 280 K, as well as energy wells that are well defined and deep. This suggests that cytochrome c maintains conformational states that are highly stable and restricted at these temperatures, which are indicative of cold adaptation in *Camelus ferus*. This stability is further substantiated by the 3D surface projections at these temperatures, as the energy wells remain steep and concentrated, indicating minimal structural fluctuations. This rigid behaviour is essential for the protein’s functionality in frigid environments with reduced kinetic energy, ensuring that the electron transport chain operates efficiently under these conditions.

At elevated temperatures, the FEL exhibits an expanding degree of conformational diversity. Multiple shallow energy minima are distributed across a broader conformational space at 303 K and 308 K, suggesting increased flexibility. This adaptability is consistent with the physiological temperatures of *Camelus dromedarius* and bolsters the protein’s functional adaptability to arid climates. Nevertheless, the FEL exhibits a highly dispersed landscape with compressed energy wells at 320 K, indicating that there is a significant sampling of a variety of conformational states and a reduction in structural stability. This is indicative of the thermal stress that occurs during conditions of extreme heat, emphasizing the limitations of cytochrome c’s adaptability. Although these structural modifications may suggest that the protein is attempting to adapt to the high temperature, the excessive flexibility implies that its functional integrity may have been compromised. The protein’s dynamic behaviour across temperatures and its adaptations to the contrasting climatic environments of *Camelus ferus* and *Camelus dromedarius* are emphasized by these findings.

### 3.5. Entropy Analysis

The entropy analysis of cytochrome c at a variety of temperatures (245 K, 280 K, 303 K, 308 K, and 320 K) offers an overview of the thermodynamic contributions of translational, rotational, and vibrational components to the overall entropy and protein dynamics. Entropy is a critical parameter in the comprehension of the thermal adaptability, structural stability, and molecular flexibility of proteins in response to a variety of thermal conditions.

Table 1 lists the entropy analysis for each component. As the thermal energy increases, the translational entropy gradually increases from 221.554 J/mol at 245 K to 227.106 J/mol at 320 K, suggesting a consistent improvement in translational freedom. In the same way, the rotational entropy exhibits a rising trend, increasing from 221.11 J/mol at 245 K to 224.9 J/mol at 308 K, and then marginally decreasing to 224.66 J/mol at 320 K. These trends are indicative of an anticipated increase in molecular mobility and overall dynamic behaviour as the temperature increases, which enhances the protein’s capacity to investigate various conformational states. The total entropy is significantly impacted by the vibrational entropy, which is subject to significant fluctuations as temperature increases. The vibrational entropy is effectively zero at 245 K and 320 K, which implies a densely packed, rigid structure and limited vibrational freedom. Conversely, the vibrational entropy considerably increases at 280 K, 303 K, and 308 K, with a peak of 13,369 J/mol at 280 K. The protein’s capacity to adapt and function efficiently in near-physiological temperatures is reflected in the high vibrational entropy, which is notably evident in *Camelus ferus* (280 K) and *Camelus dromedarius* (303 K and 308 K). This entropy is indicative of greater flexibility and dynamic motion. The total entropy is the sum of the contributions of vibrational, rotational, and translational entropy. The highest value is at 280 K (13,816.1 J/mol), and it decreases gradually as the temperature increases (12,148.4 J/mol at 303 K and 9933.69 J/mol at 308 K) before experiencing a sudden decline at 320 K (451.765 J/mol). The optimal flexibility and adaptability of cytochrome c in near-cold environments, which are characteristic of *Camelus ferus*, are underscored by the maximal entropy at 280 K. The structural destabilization and loss of vibrational freedom due to excessive thermal stress are suggested by the reduced total entropy at 320 K, despite the high translational and rotational entropy.

The thermodynamic analysis indicated low vibrational entropy at 245 K and optimal entropy contributions at 280 K, demonstrating a balance between stability and flexibility that supports the function of cytochrome c in the cold habitat of *Camelus ferus*. This behaviour aligns with previous studies highlighting the importance of structural stiffness in cold-adapted proteins for sustaining enzymatic activity under low-kinetic-energy circumstances. In contrast, cytochrome c showed more dynamic behaviour and flexibility at moderate temperatures (303 K and 308 K) with increased RMSF values, a larger FEL conformational space, and moderate SASA and Rg values. The adaptive flexibility required for efficient electron transport in *Camelus dromedarius*’s high temperatures is aligned with these alterations. At these temperatures, the protein exhibited high vibrational entropy, an adaptive feature commonly observed in thermophilic proteins. This allows the protein to absorb heat while maintaining structural integrity and functional efficiency, supporting its role in energy production under thermal stress [43,44,45].

Cytochrome c’s unique thermal adaptations are emphasized by the entropy analysis. The protein maintains a balance of rigidity and flexibility at lower temperatures (245 K and 280 K), which supports stability in cold conditions. The high vibrational entropy at moderate temperatures (303 K and 308 K) is indicative of the adaptive flexibility that is essential for the efficient operation of *Camelus dromedarius* in milder environments. Nevertheless, the thermal limit of cytochrome c’s resilience is indicated by the abrupt decrease in total entropy at 320 K, which suggests a collapse of vibrational dynamics and structural destabilization. These results offer valuable thermodynamic insights into the protein’s adaptation to divergent thermal environments. The comparative investigation of cytochrome c in *Camelus ferus* and *Camelus dromedarius* highlights the relationship between structural stability and flexibility as crucial elements in temperature adaptation [46,47]. Cold-adapted proteins often exhibit stiffness to ensure stability in low-energy conditions, whereas heat-adapted proteins depend on flexibility to manage elevated thermal energy. These molecular changes underscore the evolutionary refinement of cytochrome c for endurance in diverse temperature conditions. These findings are significant for comprehending protein adaptation in harsh environments and for future applications in protein engineering and biotechnology, where thermostable enzymes are frequently necessary for industrial processes.

## 4. Conclusions

The study investigates how cytochrome c adapts to the contrasting temperatures of the camel species’ habitats (*Camelus ferus* and *Camelus dromedarius*) using Thermal Titration Molecular Dynamics (TTMD) simulations. In the study, at reduced temperatures (245 K and 280 K), cytochrome c demonstrates remarkable structural stiffness and limited conformational variations, hence enhancing its functional stability in the frigid habitat of *Camelus ferus*. Further, at intermediate temperatures (303 K and 308 K), the protein exhibits enhanced flexibility and plasticity, indicative of the functional necessities of *Camelus dromedarius* in arid desert environments. Additionally, at elevated temperatures (320 K), cytochrome c experiences considerable structural destabilization, indicated by heightened solvent exposure, broadened conformational space, and reduced vibrational entropy. This signifies the thermal threshold beyond which cytochrome c’s structural and functional integrity is compromised. The findings highlight the relationship between structural stability and flexibility, allowing cytochrome c to adjust to varying heat conditions, thus offering insights into its evolutionary adaptation for survival in harsh climes.

## Figures and Tables

**Figure 1 animals-15-00381-f001:**
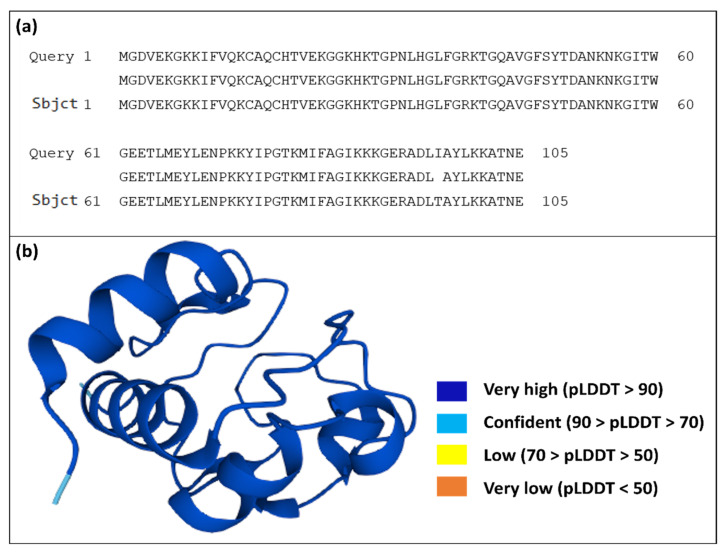
(**a**) Pairwise alignment of cytochrome c in *Camelus dromedarius* (*Arabian camel*) and *Camelus ferus* (Wild *Bactrian camel*) using BLAST; (**b**) 3D structure of cytochrome c in *Camelus dromedarius.*

**Figure 2 animals-15-00381-f002:**
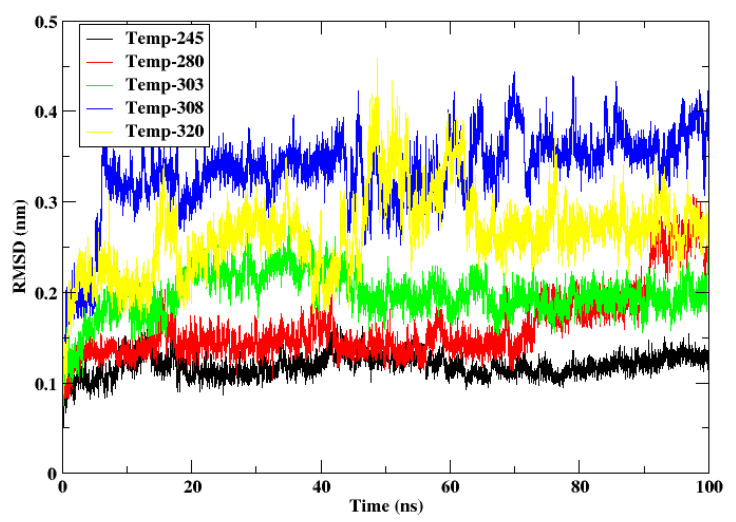
RMSD of the Cα atoms of the protein cytochrome c at the temperatures 245 K, 280 K, 303 K, 308 K, and 320 K.

**Figure 3 animals-15-00381-f003:**
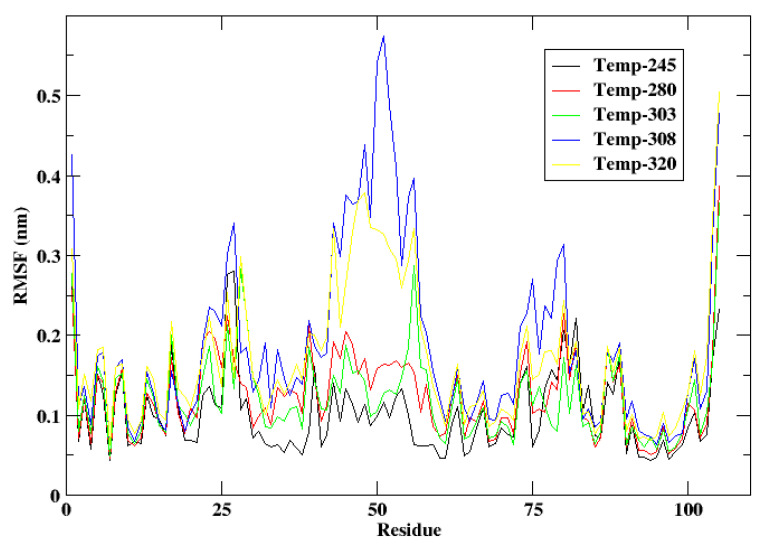
RMSF of the protein cytochrome c at the temperatures 245 K, 280 K, 303 K, 308 K, and 320 K.

**Figure 4 animals-15-00381-f004:**
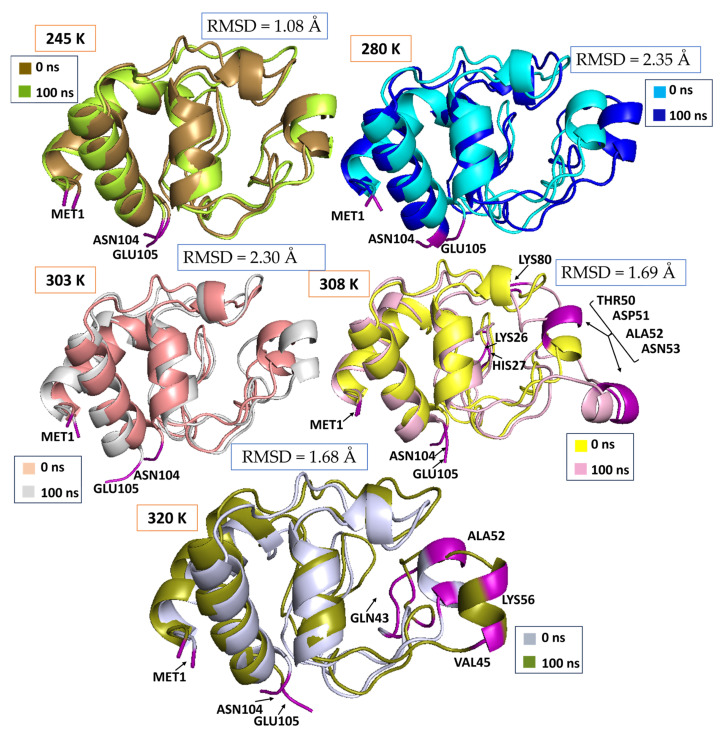
Conformation of the protein cytochrome c (0 ns and 100 ns) at the temperatures 245 K, 280 K, 303 K, 308 K, and 320 K.

**Figure 5 animals-15-00381-f005:**
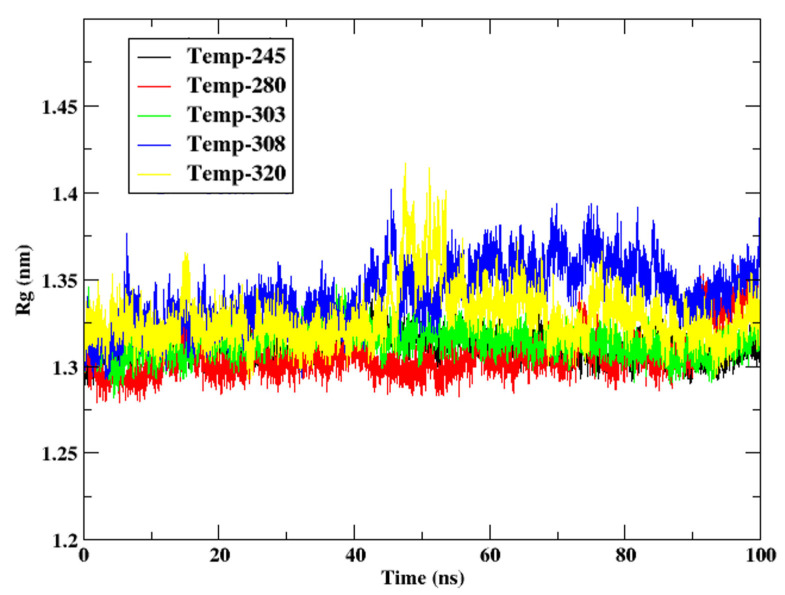
Rg of the protein cytochrome c at the temperatures 245 K, 280 K, 303 K, 308 K, and 320 K.

**Figure 6 animals-15-00381-f006:**
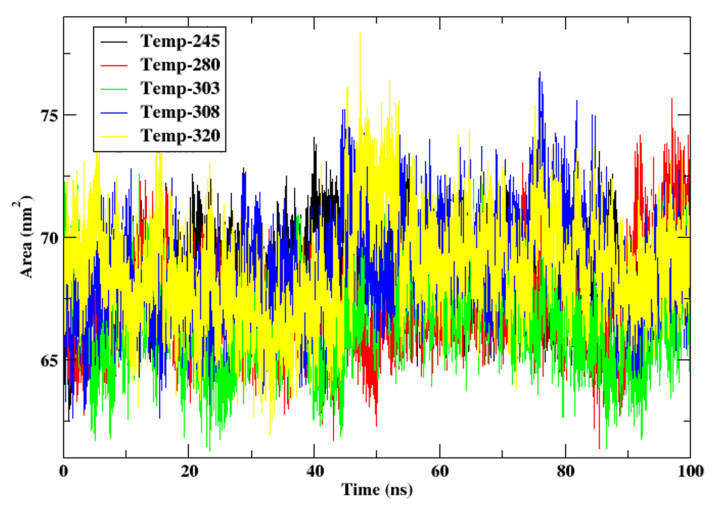
SASA of the protein cytochrome c at the temperatures 245 K, 280 K, 303 K, 308 K, and 320 K.

**Figure 7 animals-15-00381-f007:**
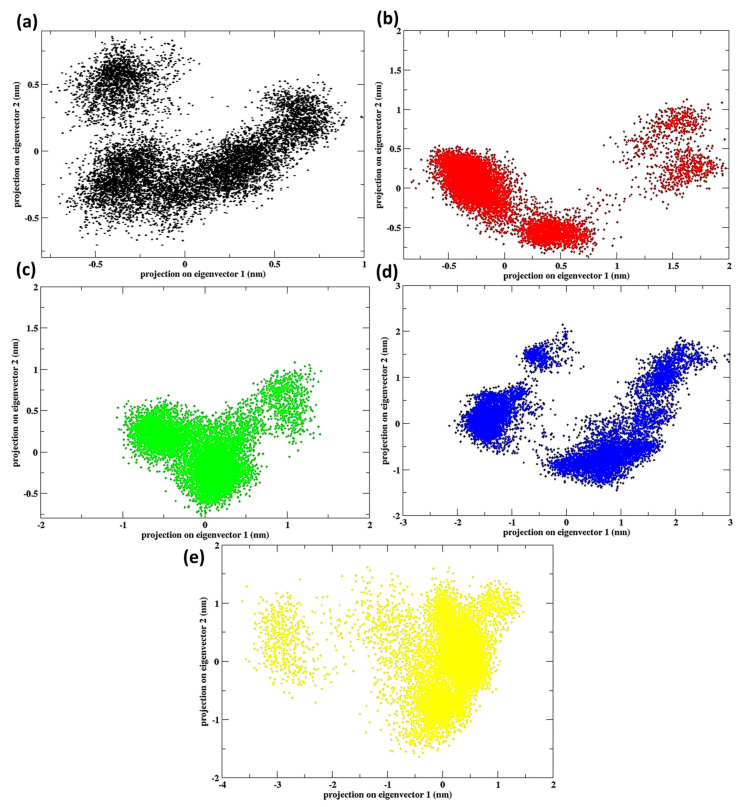
PCA of the protein cytochrome c at the temperatures (**a**) 245 K, (**b**) 280 K, (**c**) 303 K, (**d**) 308 K, and (**e**) 320 K.

**Figure 8 animals-15-00381-f008:**
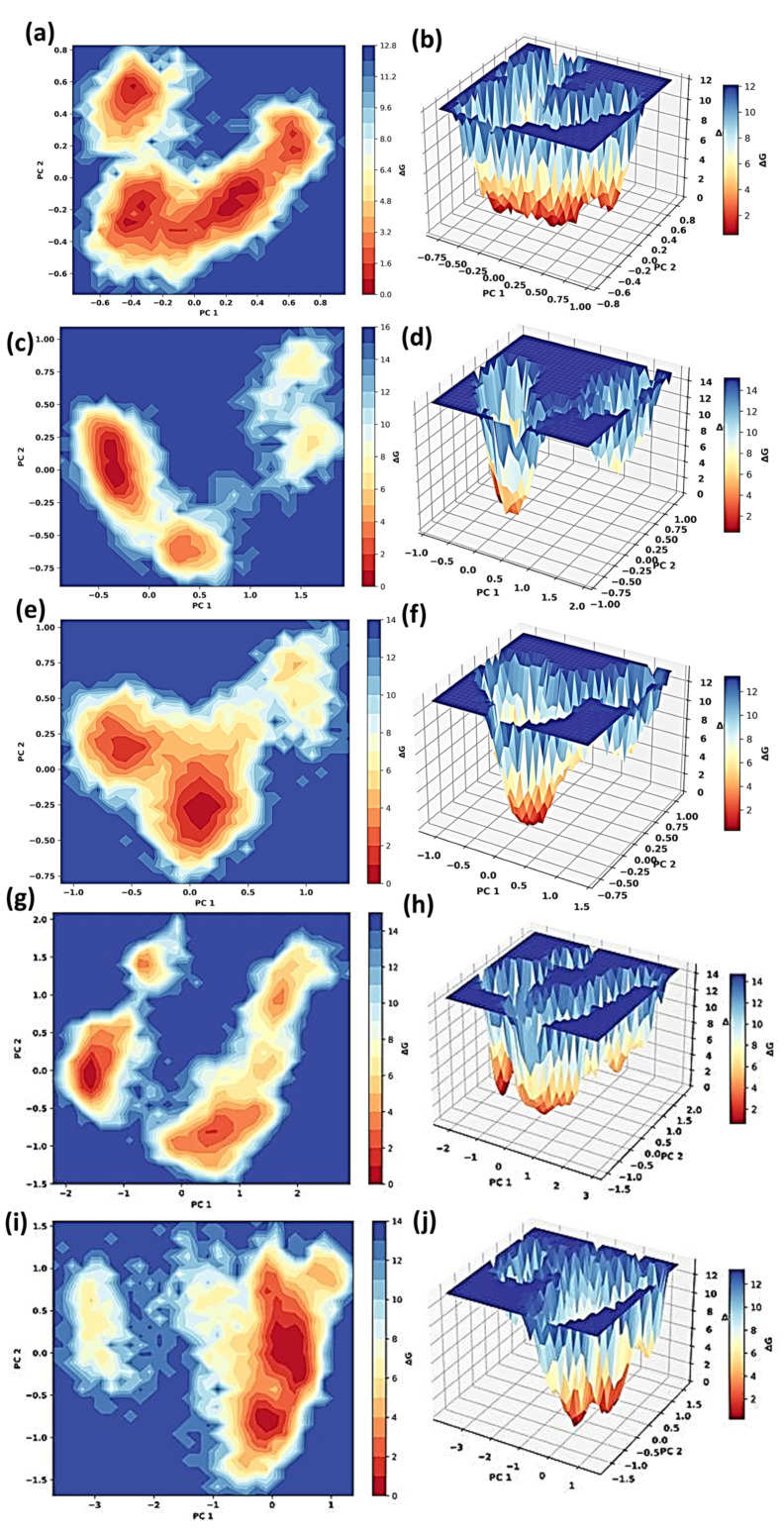
FEL of the protein cytochrome c at the temperatures (**a**,**b**) 245 K, (**c**,**d**) 280 K, (**e**,**f**) 303 K, (**g**,**h**) 308 K, and (**i**,**j**) 320 K.

**Table 1 animals-15-00381-t001:** Entropy of the protein cytochrome c at the temperatures 245 K, 280 K, 303 K, 308 K, and 320 K.

Components	245 K	280 K	303 K	308 K	320 K
Translational entropy (J/mol)	221.55	224.33	224.33	226.31	227.10
Rotational entropy (J/mol)	221.11	222.86	222.79	224.90	224.66
Vibrational entropy (J/mol)	0	13,369	11,701.30	9482.48	0
Total entropy (J/mol)	442.66	13,816.10	12,148.40	9933.69	451.76
Heat capacity (J/mol)	24.94	13,018.90	12,850.50	13,262.90	24.94
Internal energy (kJ/mol)	6.11	38,318.10	38,407.20	39,278.90	7.98
Zero-point energy (kJ/mol)	0	36,367.10	36,567.50	37,440.90	0

## Data Availability

The data are contained within the article.

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
