# Peer review of "In Silico Analysis of Temperature-Induced Structural, Stability, and Flexibility Modulations in Camel Cytochrome c"

_animals, 2025, doi:10.3390/ani15030381_

Round 1
Reviewer 1 Report
Comments and Suggestions for Authors
General comments:
The manuscript, “In Silico Analysis of Environmental Adaptation Mechanisms in Animal Enzymes: A Comparative Study of Cold and Heat-Tolerant Proteins” aimed to comprehensively investigate the characteristics of cytochrome c from Camelus ferus and Camelus dromedarius. Structural stability was assessed using a thermal titration molecular dynamic simulation and 5 temperatures ranging from 245-320 K. It was concluded that cytochrome c was the most stable at lower temperatures while flexibility increased with temperature.
I will start off by saying I like the information in this manuscript. The use of these techniques in this capacity is quite interesting. The bulk of my comments pertain to the structure of the manuscript.
This manuscript does not follow the typical format of manuscripts published in animals. Please review the instructions for authors found at https://www.mdpi.com/journal/animals/instructions. The current format makes it difficult to read as methods that are important to understand the results are split between the results and materials and methods sections. I think the authors should combine the results and discussion section because a lot of the information is discussed twice. Removing the redundancies will make the manuscript more concise and much easier to read. Please be consistent with formatting.
The title is too vague and misleading. Animal enzymes were not compared. Only cytochrome c was discussed. This should be changed to something along the line of In Silico Analysis of the Effect of Temperature On the Structure, Stability, and Flexibility of Cytochrome C from Camels.
The manuscript is missing a simple summary, and the abstract is on the longer side. All abbreviations should be defined prior to their used in the abstract, the main text, and in the figures. Spell out the section headings. There is also a need to improve figure quality so all axes are legible.
Overall, I believe the information is present in this document, but it is not organized correctly.
Line specific/Section comments.
Abstract: Please introduce all abbreviations.
Ln 27 and ln 34: This seems to be contradictory to line 34. Is this a typo? Should line 34 only say decreased stability?
Introduction: This seems to be 2 well written introductions put together to make 1? Ln 45-85 seem to be the introduction to a different manuscript perhaps. I do not see how this information pertains to the displayed data. I suggest starting the introduction on line 86 and deleting lines 45-85. Regardless, I will provide comments for it all.
Ln 63-71: This paragraph should be in past tense, and every statement begins with an anthropomorphic statement.
Ln 72-75: Where was the comparison between cytochrome c, the antifreeze proteins, and the heat shock proteins?
Ln 87 and 131: Species name format. This issue occurs elsewhere, so please double check this.
Ln 103: What elucidates the role of cytochrome c in the regulation of body temperature? The previous sentence says it is indirectly involved. I suggest deleting the sentence “This elucidates its role in the regulation of body temperature.”
ln 136: You are missing a comma after camel. "...camel, (v) ..."
Ln 86-142: This is well done. I think you do an excellent job describing your aims and introducing the temperatures used in the molecular dynamic simulations.
Results – Materials and methods should be after the introduction. The results should state the findings, not explain the methods used. For example, lines 145-148 should be in the materials and methods and not the results section. Lines 159 and 160 should be in the introduction. This section needs to be reworked.
Ln 145: This should say the sequence of cytochrome c of Camelus dromedarius was acquired from the Uniprot database. What about the sequence for Camelus ferus? Please include the ID used for this as well because there are 2 Wild Bactrian Camel sequences listed in Uniprot. Maybe you did not get the sequence from Uniprot?
Ln 147: You already have identified Camelus dromedarius and Camelus ferus as the Arabian and wild bactrian camel elsewhere. You have it defined multiple times throughout this manuscript.
Ln 149-150: Although there is only 1 AA change, threonine and isoleucine have different properties. Isoleucine is hydrophobic and nonpolar while threonine is a polar hydrophobic molecule. I think you should discuss this more especially since some of the observed changes displayed in figure 4 were in the C-terminal region. I believe you should mention the implication of change being in that region of the protein.
Ln 181-186: This is all materials and methods.
Ln 224: typo – Environments
Ln 222-226: Interpretation of results. This should be in the discussion.
Ln 231: Format species name.
Ln 250 and Figure 4: If you are showing the conformation of the protein at 0 ns of exposure to the treatment temperature, then what temperature is that conformation? Is the 0 ns conformation the conformation observed in normal physiological conditions? It appears to me they are all the same, so I am not sure why different colors were used to represent this 0 ns control. The 10 colors used imply that there are 10 variations of the protein when there are only 2: 0 ns and 100 ns at the temperature specified by the box. Please tell me if I am missing something here. Also, accessibility can be improved if you use a color that is not red to indicate differences in the protein conformation. This is hard for red/green colorblind individuals to view.
Figure 7: It is not possible to read the any of the axes.
Figure 8: It is difficult to read the axes in this figure.
Table 1: Please present the data using the same number of significant figures.
Discussion: Combine with results and remove redundancies.
Materials and Methods: The protein retrieval information answers my previous comment. Move this section after the intro, add the materials and methods information form the results and then remove the redundant information from the results.
Author Response
General comments:
The manuscript, “In Silico Analysis of Environmental Adaptation Mechanisms in Animal Enzymes: A Comparative Study of Cold and Heat-Tolerant Proteins” aimed to comprehensively investigate the characteristics of cytochrome c from Camelus ferus and Camelus dromedarius. Structural stability was assessed using a thermal titration molecular dynamic simulation and 5 temperatures ranging from 245-320 K. It was concluded that cytochrome c was the most stable at lower temperatures while flexibility increased with temperature.
I will start off by saying I like the information in this manuscript. The use of these techniques in this capacity is quite interesting. The bulk of my comments pertain to the structure of the manuscript.
Comment 1.
This manuscript does not follow the typical format of manuscripts published in animals. Please review the instructions for authors found at https://www.mdpi.com/journal/animals/instructions. The current format makes it difficult to read as methods that are important to understand the results are split between the results and materials and methods sections. I think the authors should combine the results and discussion section because a lot of the information is discussed twice. Removing the redundancies will make the manuscript more concise and much easier to read. Please be consistent with formatting.
Response: Thank you for the comments. The manuscript has been formatted according to the manuscripts published in animals. As suggested by the reviewer, the results and discussion section has been combined to remove redundancies. Kindly review the revised manuscript.
Comment 2.
The title is too vague and misleading. Animal enzymes were not compared. Only cytochrome c was discussed. This should be changed to something along the line of In Silico Analysis of the Effect of Temperature On the Structure, Stability, and Flexibility of Cytochrome C from Camels.
Response: As suggested by the reviewer, the title has been modified accordingly. Kindly review the revised manuscript.
Comment 3.
The manuscript is missing a simple summary, and the abstract is on the longer side. All abbreviations should be defined prior to their used in the abstract, the main text, and in the figures. Spell out the section headings. There is also a need to improve figure quality so all axes are legible.
Overall, I believe the information is present in this document, but it is not organized correctly.
Response: The simple summary has been added and the abstract has been shortened. Kindly review the “Simple summary” and “Abstract” section of the revised manuscript.
According to the format of published manuscript in animals, the Abbreviations has been added in the last section. However, all the abbreviation has been defined in the abstract, the main text, and in the figures accordingly. The section headings were spell out and the Figures were improved accordingly. Kindly review the revised manuscript.
Line specific/Section comments.
Comment 4.
Abstract: Please introduce all abbreviations.
Ln 27 and ln 34: This seems to be contradictory to line 34. Is this a typo? Should line 34 only say decreased stability?
Response: All the abbreviations has been introduced in the abstract and the typo sentence in line 34 has been modified. Kindly review the “Abstract” section of the revised manuscript.
Comment 5.
Introduction: This seems to be 2 well written introductions put together to make 1? Ln 45-85 seem to be the introduction to a different manuscript perhaps. I do not see how this information pertains to the displayed data. I suggest starting the introduction on line 86 and deleting lines 45-85. Regardless, I will provide comments for it all.
Ln 63-71: This paragraph should be in past tense, and every statement begins with an anthropomorphic statement.
Ln 72-75: Where was the comparison between cytochrome c, the antifreeze proteins, and the heat shock proteins?
Response: As suggested by the reviewer, the introduction was started from line 86 and lines 45-85 were deleted. Kindly review the “Introduction” section of the revised manuscript.
Comment 6.
Ln 87 and 131: Species name format. This issue occurs elsewhere, so please double check this. (Line )
Ln 103: What elucidates the role of cytochrome c in the regulation of body temperature? The previous sentence says it is indirectly involved. I suggest deleting the sentence “This elucidates its role in the regulation of body temperature.”
ln 136: You are missing a comma after camel. "...camel, (v) ..."
Ln 86-142: This is well done. I think you do an excellent job describing your aims and introducing the temperatures used in the molecular dynamic simulations.
Response: Thank you for your comment. The species name format has been corrected. the sentence “This elucidates its role in the regulation of body temperature.” has been deleted. The comma after camel. "...camel, (v) ..." has been added. Kindly review the highlighted portions of the “Introduction” section of the revised manuscript.
Comment 7.
Results – Materials and methods should be after the introduction. The results should state the findings, not explain the methods used. For example, lines 145-148 should be in the materials and methods and not the results section. Lines 159 and 160 should be in the introduction. This section needs to be reworked.
Response: The Results section was reworked as suggested by the reviewer. The Materials and methods were put after the introduction section. The lines 145-148 were removed and added in the result section. Lines 159 and 160 was moved to the introduction section. Kindly review “ Results and Discussion”, “Materials and Methods” and “Introduction” section of the revised manuscript.
Ln 145: This should say the sequence of cytochrome c of Camelus dromedarius was acquired from the Uniprot database. What about the sequence for Camelus ferus? Please include the ID used for this as well because there are 2 Wild Bactrian Camel sequences listed in Uniprot. Maybe you did not get the sequence from Uniprot?
Response: The ID used for the sequence of Camelus ferus was added in the Materials and methods section. Kindly review the “2.1. Protein Retrieval” section of the revised manuscript.
Ln 147: You already have identified Camelus dromedarius and Camelus ferus as the Arabian and wild bactrian camel elsewhere. You have it defined multiple times throughout this manuscript.
Response: We have identified Camelus dromedarius and Camelus ferus as the Arabian and wild bactrian camel in the Introduction section. We removed it from the Results section and Materials and methods section. Kindly review the revised manuscript.
Ln 149-150: Although there is only 1 AA change, threonine and isoleucine have different properties. Isoleucine is hydrophobic and nonpolar while threonine is a polar hydrophobic molecule. I think you should discuss this more especially since some of the observed changes displayed in figure 4 were in the C-terminal region. I believe you should mention the implication of change being in that region of the protein.
Response: As suggested by the reviewer, we mention the implication of change being in that region of the protein. Kindly review the “3.1. Protein Structure” section of the revised manuscript.
Ln 181-186: This is all materials and methods.
Response: We have moved it to the materials and methods section. Kindly review the materials and methods section of the revised manuscript.
Ln 224: typo – Environments
Response: The typo has been corrected. Kindly review the revised manuscript.
Ln 222-226: Interpretation of results. This should be in the discussion. (234)
Response: We have combined the Discussion with results section as suggested earlier. Kindly review the “Results and Discussion” section of the revised manuscript.
Ln 231: Format species name.(243)
Response: Format species name has been corrected. Kindly review the “3. Results and Discussion” section of the revised manuscript.
Ln 250 and Figure 4: If you are showing the conformation of the protein at 0 ns of exposure to the treatment temperature, then what temperature is that conformation? Is the 0 ns conformation the conformation observed in normal physiological conditions? It appears to me they are all the same, so I am not sure why different colors were used to represent this 0 ns control. The 10 colors used imply that there are 10 variations of the protein when there are only 2: 0 ns and 100 ns at the temperature specified by the box. Please tell me if I am missing something here. Also, accessibility can be improved if you use a color that is not red to indicate differences in the protein conformation. This is hard for red/green colorblind individuals to view.
Response: The conformations of the protein at 0 ns were taken at different temperatures (245 K, 280 K, 303 K, 308 K, and 320 K), not under normal physiological conditions. Each 0 ns conformation corresponds to the initial state of the protein at the specific temperature. The 10 colors represent the conformations of the protein at both 0 ns and 100 ns for the five temperatures (245 K, 280 K, 303 K, 308 K, and 320 K). The 0 ns and 100 ns conformations are distinct for each temperature, which is why different colors were used.
To clarify how different the structures were from each other the conformation changes between 0 ns and 100 ns were analyzed using RMSD (Root Mean Square Deviation) values, which are provided in the figure. These values quantify the deviation between the initial and final states of the protein at each temperature. Further, we have adjusted the color scheme to improve visibility for individuals with red-green colorblindness. The revised colors ensure that the conformational differences are distinguishable to all viewers.
Kindly review the “3.2.3. Conformation” section of the revised manuscript.
Figure 7: It is not possible to read the any of the axes.
Figure 8: It is difficult to read the axes in this figure.
Table 1: Please present the data using the same number of significant figures.
Response: We have refined the Figure 7 and 8. The Table was presented with the same number of significant figures. Kindly review the Figure 7, 8 and Table 1 of the revised manuscript.
Comment 7.
Discussion: Combine with results and remove redundancies.
Response: The discussion was combined with the results section to remove the redundancies. Kindly review the “3. Results and Discussion” section of the revised manuscript.
Comment 8.
Materials and Methods: The protein retrieval information answers my previous comment. Move this section after the intro, add the materials and methods information form the results and then remove the redundant information from the results.
Response: As suggested by the reviewer, the materials and methods section was modified accordingly. Kindly review the materials and methods section of the revised manuscript.
Reviewer 2 Report
Comments and Suggestions for Authors
See the attached file.

Author Response
animals-3392705-peer-review-report-v1.
Thank you for the opportunity to review this paper. It is satisfactorily written but not suitable for publication in its present state and, therefore, requires revision. I have highlighted areas that require attention and revision in all the sections. Addressing these points would make the paper more comprehensive, providing better context and interpretation for readers unfamiliar with the study.
Abstract
Comment 1. The opening sentence could be more engaging and specific. Rather than simply stating the importance of cytochrome c in the electron transport chain, consider briefly mentioning its role in the context of temperature adaptation for the two species.
Response: Thank you for the comment. The abstract has been modified accordingly. Kindly review the “Abstract” section of the revised manuscript.
Comment 2. Some readers might be unfamiliar with the term ‘Thermal Titration Molecular Dynamics (TTMD) simulation.’ As it's a specialized term, it could be beneficial to briefly define or explain the acronym at least once.
Response: The acronym has explained. Kindly review the “Abstract” section of the revised manuscript.
Comment 3. Mentioning the type of simulation software and the duration of the simulation could give more insight into the reliability and scope of the findings.
Response: The simulation software and the duration of the simulation has been mentioned. Kindly review the “Abstract” section of the revised manuscript.
Comment 4. "Energy wells were well characterised" could be improved to avoid repetition. A more concise phrasing like "Energy wells were well-defined" would be preferable.
Response: The abstract has been modified accordingly. Kindly review the “Abstract” section of the revised manuscript.
Introduction
Comment 5. Lines 47-49: Correct the sentence. Suggestion: Animals that live in harsh conditions, like deserts or freezing polar regions, must have specialised biochemical and molecular adaptations to survive and grow [1].
Response: The Introduction has been modified to make it more concise. The sentences have been corrected accordingly. Kindly review the “Introduction” section of revised manuscript.
Comment 6. Lines 95-97: Recast for clarity. Suggestion: By examining cytochrome c in these camels, the study seeks to comprehend the structural and dynamic adaptations of this vital protein that enable its survival and efficient function under disparate heat conditions [15].
Response: The sentence has been modified accordingly. Kindly review the revised manuscript.
Comment 7. The introduction is lengthy and sometimes repetitive. For instance, the role of cytochrome c in energy metabolism, thermoregulation, and heat generation is mentioned several times in slightly different contexts. Condensing these explanations could make the introduction more concise.
Response: The Introduction has been modified accordingly. Kindly review the “Introduction” section of the revised manuscript.
Comment 8. The discussion of cytochrome c's function in thermoregulation and its roles in both energy production and apoptosis is spread across multiple sentences. Grouping related ideas together will help improve readability.
Response: The sentences has been modified accordingly. Kindly review the “Introduction” section of the revised manuscript.
Comment 9. The introduction references previous research but doesn't clearly highlight the necessity or uniqueness of the current study. Stating the specific gap in literature addressed by this research, such as the adaptation of cytochrome c in camels, would make the introduction more focused and compelling.
Response: The Introduction has been modified accordingly. Kindly review the “Introduction” section of the revised manuscript.
Comment 10. Some details, like the mechanisms of ATP production in brown adipose tissue or cytochrome c’s role in apoptosis, may be too technical for the introduction. Simplifying or saving them for the discussion section would help maintain focus and clarity in the introduction.
Response: The technical parts in the introduction has been simplified. Kindly review the “Introduction” section of the revised manuscript.
Comment 11. The introduction’s conclusion, which introduces the study’s aims, is somewhat hidden in the detailed discussion of camel environments. Reorganizing it to clearly state the study’s goals, such as investigating cytochrome c’s responses in camels using MD simulations, would strengthen the conclusion.
Response: We have reorganized the introduction’s conclusion. Kindly review the “Introduction” section of the revised manuscript.
Discussion
Comment 12. The discussion spends a lot of time detailing specific data points, such as low RMSD, restricted RMSF, and energy minima, which could be overwhelming for readers. While these results are important, they could be summarized more succinctly, with a focus on the broader implications of these findings.
Response: The discussion has been combined with the results section for simplification and removing redundancies. Kindly review the “Results and Discussion” section of the revised manuscript.
Comment 13. The section jumps between different temperature conditions (cold, moderate, and high) and molecular properties (RMSD, RMSF, entropy), which can make the discussion feel somewhat fragmented. Consider structuring the paragraph to focus first on cold conditions, then on moderate, and finally on high temperatures, while summarizing the key findings for each phase before transitioning to the next.
Response: The structuring of the paragraphs were modified accordingly. Kindly review the “Results and Discussion” section of the revised manuscript.
Comment 14. Phrases like "the protein could absorb heat without losing any of its functional efficiency" are mentioned and repeated, but this concept is also covered in the earlier part of the paragraph about "thermophilic proteins" and their adaptations. Combining these ideas into a single, clear explanation would strengthen the argument.
Response: The phrases have been modified accordingly. Kindly review the revised manuscript.
Comment 15. There are minor stylistic issues, such as "in line with other studies" and "as seen by higher RMSD" that could be more fluidly written. Instead, you might say, "consistent with other studies" or "as indicated by higher RMSD."
Response: The sentences has been modified accordingly. Kindly review the revised manuscript.
Materials and Methods
Comment 16. Lines 574-576: The temperatures 303K and 303K are repeated twice in the list of temperatures in the NVT ensemble. This redundancy should be corrected, as it is unclear why the temperature would appear twice, especially when a different temperature is expected to be used. This repetition could confuse the reader about the experimental conditions.
Response: The sentence has been corrected. Kindly review the “2.2. Thermal Titration Molecular Dynamics” section of the revised manuscript.
Comment 17. Lines 576-577: Recast this sentence to enhance clarity and comprehension. Suggestion: Throughout the 100 ns production cycle, the structure coordinates were recorded every 10 ps.
Response: As suggested by the reviewer, the sentence has been refined. Kindly review the “2.2. Thermal Titration Molecular Dynamics” section of the revised manuscript.
Comment 18. The section uses both "TTMD" and "Molecular Dynamics (MD)" but doesn't clearly define the relationship between the two. It could be helpful to clarify whether TTMD is a specific subset of MD simulations or a distinct method. The use of these terms might confuse readers who are not familiar with the nuances of molecular dynamics simulations.
Response: As suggested by the reviewer, we have only used the acronym "TTMD" throughout the manuscript. The full explanation of TTMD as a temperature-focused application is provided in the “2.2. Thermal Titration Molecular Dynamics” section of the revised manuscript. Kindly review the revised manuscript.
Discussion
Comment 19. Line 590-591: This sentence, "The regulation was established to demonstrate the progression of personal computers", is unclear. It appears to be a typographical error or misstatement, as "personal computers" are not relevant here. The intention might have been to explain how the PCA process was regulated, but the sentence needs to be rephrased for clarity.
Response: As suggested by the reviewer, the sentence has been refined. Kindly review the “2.3. Principal Component Analysis” section of the revised manuscript.
Comment 20. The paper references several tools, such as GROMACS 2022.4, PDB2GMX, and others, but lacks sufficient detail on the specific parameters or settings used in these software packages (e.g., force fields, cutoff values, and temperature control methods). A more comprehensive description of the parameters would be beneficial, as different settings can significantly influence simulation results.
Response: A detailed list of the parameters was provided in the revised manuscript. Kindly review the “2.2. Thermal Titration Molecular Dynamics” section of the revised manuscript.
Comment 21. The acronym "GROMACS" is listed four times, but its use is inconsistent. To maintain clarity, define it and ensure consistent formatting and capitalization across the text.
Response: As suggested by the reviewer, we have refined the usage of the acronym "GROMACS" for clarity and consistency. The full name, "GROningen MAchine for Chemical Simulations," is introduced upon its first mention in the “2.2. Thermal Titration Molecular Dynamics” section. In subsequent mentions, the acronym "GROMACS" has been used consistently throughout the manuscript to ensure readability. Kindly review the revised manuscript.
Conclusion
Comment 22. Line 610-611: The phrase "protein adapts to different temperatures typical of both habitats" is somewhat redundant. The context clearly shows that the study focuses on temperature adaptation, so this could be rephrased to avoid repetition. For example, "The study investigates how cytochrome c adapts to the contrasting temperatures of the camel species' habitats."
Response: As suggested by the reviewer, the sentence has been refined. Kindly review the “Conclusion” section of the revised manuscript.
Comment 23. Line 619-620: The phrase "This signifies the heat threshold of cytochrome c's structural and functional integrity" could be more precisely phrased as "This signifies the thermal threshold beyond which cytochrome c’s structural and functional integrity is compromised."
Response: As suggested by the reviewer, the sentence has been refined. Kindly review the “Conclusion” section of the revised manuscript.
Round 2
Reviewer 1 Report
Comments and Suggestions for Authors
Thank you for updating the manuscript and responding to the provided comments. The current format makes it a lot easier to read. Thank you for updating the figures and including the text from 314-326; this section was very helpful. Your clarification as to why 0 ns differed between protein conditions on figure 4 was helpful and I agree that different colors are needed to denote that each protein started at a different conformation. I think you did an excellent job with this revision.